# "Pro Honore et Libertate Ecclesiae Invicta Fortitude Sustinuit"—The Oratory of St Thomas Becket in the Cathedral of Anagni

**Claudia Quattrocchi**

Ministry of Culture (Italy), National Central Library of Rome, Viale Castro Pretorio, 105, 00100 Roma, Italy; claudia.quattrocchi@beniculturali.it or claudiaquattrocchi4@gmail.com

**Abstract:** On the 9th of October, 1170 Pope Alexander III resided in Anagni, which had been the ancient residence of the court of the Popes for at least two centuries. He wrote to two influential local archbishops for help in pacifying King Henry II and Archbishop Thomas Becket, who had been in dispute for six years. Sensing Becket's looming tragic fate, Alexander III began slowly to encircle the archbishop with rhetoric of the new martyr of *Libertas Ecclesiae*. When he had to flee from Rome besieged by factions led by Frederick I, the pope found refuge in Segni, where he canonised Thomas Becket on 21 February 1173. However, it was in faithful Anagni that he settled on and off from March 1173 through the following years (November 1176; December 1177–March 1178; September 1179). It was here that he decided to elaborate a powerful speech in images. In an oratory in the crypt of the grandiose cathedral, Alexander III had the last painful moments of the Archbishop's death painted in a program imitating that of St. Peter's in the Vatican. Becket thus became the new imitator of Christ, the new Peter, the new martyr on the altar of the Church of Rome.

**Keywords:** Anagni; Canterbury; Saint Thomas Becket; medieval wall painting; papacy





## 1. Introduction

The Cathedral of Anagni is its most famous medieval monument (Figure 1a), and the Oratory of St. Thomas Becket is the second most prominent chamber in the levels of its crypt. The stairs that lead to this chamber also give access to the main hall crypt, dedicated to St. Magnus (San Magno), martyr and bishop, and others saints.

The staircase, remodelled in the 17th century, is set at the third bay of the hall crypt's left nave[1]; it terminates in a small room commonly referred to as the hall which gives access to the two crypt chambers (Figure 1b, no. 8 on plan).

The crypt of St. Magnus, which is contextual to the cathedral in its architecture, holds intact an example of total decoration. With its 540 square metres of frescoes, the paintings represent the History of the Salvation from before Time to its final apocalyptic Resolution. In this view, the exaltation of the Priest over the King is central. Salvation is only achieved in the Church that will be destined to rule on Earth. The main strands in which the visual argumentation is articulated always bring the example of the 'supremacy of the Priest' from the biblical one, to Christ, to the example of Saint Magnus. The entire room was decorated between the 12th and late 13th century by three workshops: the so-called Three Masters of Anagni. The theme of the priesthood according to Christ and the martyrdom of the bishop are subjects that fit in well with the later decoration of Becket's oratory. In addition to the level on which they extend, the underlying theme, the style also links the two environments. With the exception of a small part, the paintings in the Oratory are generally assigned to a pupil of the First Master who was active on 70% of the surfaces of the main crypt.

The chamber that now serves as the Oratory of St. Thomas Becket (henceforth, Oratory) occupies a large part of the south-west side of the cathedral. The chamber had been

previously frequented in medieval times. The current medieval decoration, dated between the 12th and 13th centuries, extends over 180 square meters. The paintings, not in a good state of preservation, present subjects from the Old and New Testament arranged on all available surfaces following the models of the early Christian Roman basilicas decorated with biblical narrative cycles (Figure 2). Among its decoration are four episodes from the last moments of the life of the saint and a portrait of Becket between Christ, the virgin and other saints.

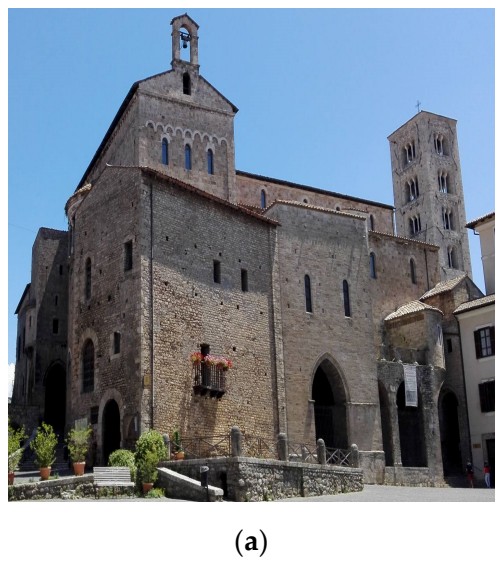

(**a**)

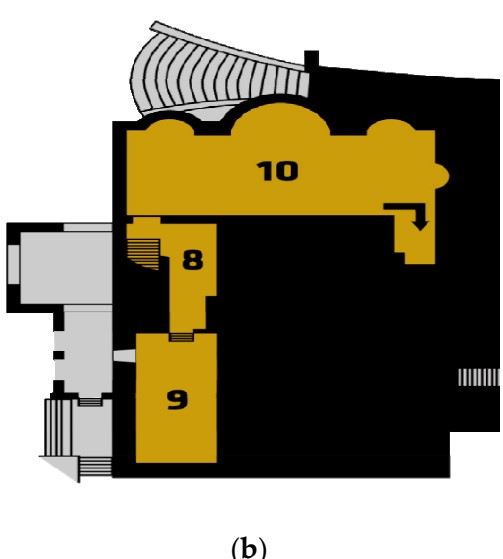

(**b**)

**Figure 1.** Anagni, Cathedral (**a**) southwest side view ©author; (**b**) plan of the rooms on the level of the crypts. 8. Vestibule; 9. Oratory of St. Thomas Becket; 10. Crypt of St Magnus ©author.

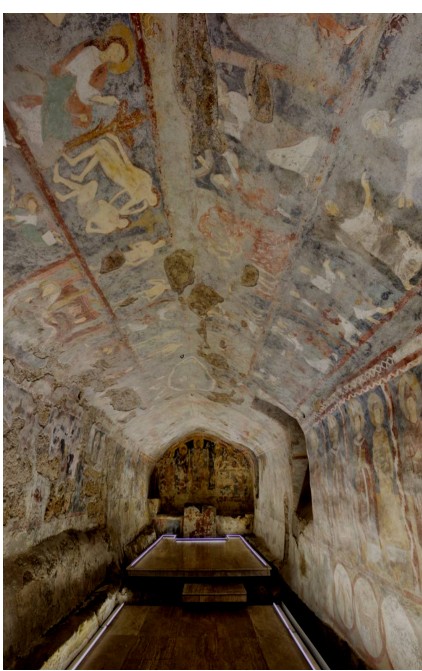

**Figure 2.** Anagni, Cathedral. Oratory of St. Thomas Becket. Vision of the interior © Museo della Cattedrale di Anagni (with kind permission).

The aim of this essay is to reconstruct the critical aspect of the oratory from various points of view—the archaeological, architectural and pictorial points of view—according to the latest research. In fact, on the occasion of the 850th anniversary of the death of St

Thomas Becket, there has been a proliferation of studies on this subject[2]. In light of more or less recent interventions, it will be demonstrated that we are here dealing with the first full-length portrait of Saint Thomas Becket in episcopal robes (the first to have survived); the paintings date from the aftermath of Alexander III's sanctification in Segni (a town opposite Anagni); they were commissioned by the pope himself during his stay (1173–1176) to draw up a new and powerful manifesto in favour of the Church of Rome against the Empire of Frederick I.

To understand the presence of this artistic testimony in a place seemingly so distant from Canterbury, it is necessary to focus on the importance of Anagni from the end of the 12th century, and of the cathedral as a setting for the events of the Papacy.

Anagni dominates the ancient via Latina approximately 60 km from Rome. Since the middle of the 1000s, it had been an increasingly frequent destination of the papal itinerary. When Rome was unsafe, the Curia repaired to other locations, usually nearby, from which it would be easy to rule and to return to Rome. Anagni became a favourite point and a compulsory stop when the Curia was in motion. The city soon became a truly international papal city, a trusted territorial base for the exercise of temporal power over the area of the *Patrimonium Sancti Petri*.

When Anagni became one of the seats from which Pope Alexander III governed the Church, the city itself took on an international status. This is true both with regard to Alexander III's relations with Frederick I and with regard to the diplomatic management of the 'English affair'. The Cathedral of Anagni, founded in its present form by Bishop Pietro da Salerno between 1164 and 1105, is one of the most majestic of the Italian Romanesque monuments. It is the site of central events in European history, which it may be useful to recall in order to better understand the importance of the historical context.

On 1 September 1159, Pope Adrian IV, the only Englishman to sit on the Throne of Peter, passed away in Anagni. His last words were to beseech the Church of Canterbury for charity toward his elderly mother. Cardinal Bandinelli, the former Chancellor of the Curia, sought to bury the pope in the cathedral of Anagni and to convene the Conclave for the succession there as well. It is evident, however, that already by this time Anagni Cathedral had come to be understood as a papal basilica, loyal to Rome. The imperial party of the Curia, however, succeeded in having the Conclave celebrated in Rome. The result was a Schism that opened immediately and lasted until 1178: for all but the last three years of his papacy, Alexander III was opposed by Victor IV. As is well known, Alexander's pontificate was consequently spent almost entirely away from Rome.

On 24 March 1160, following the infamous Council of Pavia in which Alexander was pronounced antipope by the followers of Victor, Alexander returns the favour: from the high altar of Anagni Cathedral, he excommunicates Frederick I and his antipope Victor IV. In September 1161, Anagni Cathedral witnessed an event of enormous importance in the relations between Anagni and English history. Alexander received the legates, sent by Henry II, presented the pope with a dossier pleading the case for sanctity of King Edward. From the same altar at which, a year earlier, he had excommunicated Frederick I, Alexander elevated King Edward to the rank of Holy Confessor.

Referring to the fateful year 1170, at the beginning of October, he goes up the Via Latina, passing through Alatri on his way to Ferentino (near Anagni), where he is attested on 16 September. He spends from 8th to 10th of October in Anagni, and then by 17th of October he was in Tusculum, where he remained until early March of 1173, engaged in the defence of the city which had been claimed by the Roman Commune. The defeat of the Pope's army was total and the city was razed to its foundations. It is amid this scene of physical and political rubble that Alexander learns of Becket's death (Ambrosi De Magistris 1889, II, p. 144). Once again, Alexander had to retreat south.

Defeated on several fronts, the Pope needed to find a new rhetorical impetus for his personal affirmation. Already convinced by the dramatic nature of Becket's murder, by the pressure of public opinion, and by the accounts of eyewitnesses, as soon as he arrived in Segni in February 1173, the pontiff elevated Becket to the rank of saint. From 27 March to 8

October 1173 (Jaffé 1888, II, pp. 265, 277), Alexander was once again resident in Anagni. On 12 March of that year, official ratification of Becket's cult was transmitted to the Canterbury chapter, which responded by testifying to numerous miracles already attributed to Becket's body. A few days after the canonisation, Alexander became the first pontiff 'to express the idea of the intercession of the martyr in the remission of sins', instituting a 'Becket Jubilee' for all those who went to honour the saint's tomb in Canterbury.

It must have been precisely at this moment that Alexander decided to accompany his valorisation of the cult of Becket with an act of patronage.

## 2. Materials and Results

The impetus for my investigation (Quattrocchi 2017) was the paradox of having a large amount of restoration documents that had neither been published (ISCR 1987–2003) nor interpreted in light of the broader contexts of the cathedral or of historical and artistic paradigms.[3] Other factors have contributed to the scarcity of studies. The first is accessibility. Due to restorations and lack of musealisation, from the mid-1980s until 2015, the chamber was only visible to scholars. Secondly, the poor condition of the painted surface makes it difficult to read the scenes and the overall quality of the surviving images is low. Even in the face of the clear historical and artistic importance of the Oratory, studies have always been limited to a few citations (see Toesca 1996; van Marle 1932, I, p. 192; Hermanin 1945, p. 264; Matthiae and Gandolfo 1988, pp. 133–34; Hugenholtz [1979] 2001, pp. 47–69; Boskovits 1979, pp. 3–41; Parlato and Romano 2001, pp. 254–55; Kessler 1989, pp. 132–35; Kessler 2002, pp. 59–61). The first significant analysis was undertaken by Kessler (Kessler 2001, pp. 93–103; Kessler 2002, pp. 141–57), who dates the paintings between 1173 and no later than the first quarter of the 13th century. The next was by Moretti (Moretti 2008, 2010), who identifies archbishop Stephen of Langton as the *concepteur* of the paintings, which she dates to the age of Innocent III. A few years later, Cipollaro and Decker proposed Kessler's chronology of the late 12th to the middle of the 13th century. Although they realised that there were several decorative phases, they did not have documentation of the restorations. This makes them uncertain whether Becket's scenes are contextual to the decoration (Cipollaro and Decker 2013). In 2017, I combined a critical reading of the unpublished results of the restoration with a first complete study of the Oratory from an archaeological and architectural perspective, including a complete mapping of the paintings. The result was a more precise historical reading, which attributed the conception of the cycle and its realisation primarily to the will of Pope Alexander III, enacted over a period from March 1173 to 1176 at the latest. The aim of this contribution is to summarise the results of that research and to update them in light of the most recent reflections and studies on the cult and other artistic evidence associated with St Thomas Becekt.

## 3. The Method

In the analysis of the chamber, three fundamental aspects will be taken into account, investigated according to different integrated methodologies. Firstly, the reconstruction of the material aspect of the crypt will be reported on, followed by the archaeological and architectural history of the oratory. For the section on topographical, archaeological and architectural analysis, the results of the archaeological excavations carried out and the metrological analysis of the walls will be used. As for the reconstruction of the pictorial phases, many restoration documents and historical photographs are available and will be supplemented with the results of the de visu analysis and artistic techniques. Finally, the iconographic analysis.

Then, this article turns to a discussion of the main subjects comprising the cycle, with particular attention to those scenes in which St. Thomas Becket appears. Kessler reminds us that the painted programme, as conceived, denotes "a desire to associate the oratory with the Church of Rome" (2001, p. 93). In particular, he discerns that "the reference to Rome is even more evident in the nave of the chapel, where the cycles of the Old and New Testaments recall the frescoes of Old St Peter's, both in the whole programme and in the

single episodes" (Kessler 2001, p. 97). This current study of the iconographic programme of the oratory has therefore followed two different and complementary methodological paths. The first approach is a "transversal" one, in which the scenes are analysed in their roles as "copies", in relation both to "late antique Roman models" and to other more or less contemporary "medieval copies of Old St. Peter's". The second approach is "monographic," in which the painting are analysed as discrete creations for the Oratory at Anagni. Although these critical perspectives are interdependent, a "vertical" type of investigation is proposed here. The aim is to provide as complete an image as possible of the environment of the Oratory and of the cultural value of the artistic testimony it preserves. Nevertheless, two historiographic categories remain at the core of this work: "centre-periphery," with Rome as "centre"; and "model-copy," with St Peter's, St Paul's outside the walls and the late antique basilicas as models (see Spieser 2020, pp. 27–46; Romano 2020, pp. 127–41).

## 4. The Pre-Existence Hypothesis. New Definition of the Architectural Phases

The following section has three objectives. The first is to disprove the ancient idea that this oratory is a Roman structure once used as a mithraeum. The second objective will then be to scientifically investigate the hypothesis of the pre-existence of this chamber on the basis of archaeological research and analysis of the walls. Once this is done, it will then be possible to define how the oratory was defined in its current volumetry and form. It should be noted that in its architectural appearance, the oratory is completely 'in phase' with the Romanesque cathedral. It existed as it is visible today at the time of the consecration of the cathedral in 1105. If one is looking for ancient and early medieval pre-existences, they are to be found in the level below the floor level or in the re-use of some boulders for the septum of the final wall. It is therefore necessary to begin with a brief discussion of the building history of the present cathedral, the forms of which were largely determined by the Romanesque construction workshop.

After his miraculous discovery of the relics of St. Magnus, Bishop Peter of Salerno (1064–1105) decided to raze the early medieval cathedral. Building commenced in 1072 and was completed in 1105, a huge undertaking in terms of commitment and costs and a sign of the papal transformation of Anagni. The city was a nodal point on the Rome-Montecassino axis and, together with nearby Segni, it was part of the true 'vanguard' of the Gregorian party.[4] The church is oriented on a north–south axis and comprises a nave and two aisles, a triapsidal presbytery located to the northwest, and a minimally projecting transept. The simple basilica façade is divided into three parts by portals of the Campanian type, while the side aisles and clerestory are continuous and slightly articulated with simple single-light windows. The bell tower is separate from the church, set forward and slightly off-axis with respect to the façade. The almost total concordance of modular proportions (from the length to the portals) has led to talk of "derivation or filiation" from the abbey church of Montecassino (1066–1071) (Urcioli 2006, p. 191), and even to a hypothesis of "the possible use in both buildings of the same workers or, at least, of common construction directives" (Urcioli 2006, p. 193).

A second intervention, in the early Gothic period, involved modifications to both the exterior and the interior of the cathedral. On the outside, this phase is associated with the addition of an archway with animal protomes on pendentives (now visible beneath the portico facing of the piazza, on the side of the Oratory). Inside, the system of supports in the upper basilica was altered from a system of pillars alternating with columns, in which the former presented a rectangular form (with the longer side along the nave), to one of cruciform pillars. Urciuoli assigns these changes to the years around 1179, when we encounter a report of the reconsecration of the cathedral by Pope Alexander III (Urcioli 2006, p. 199). Let us stop at this date and the cathedral's appearance at this time, which was in fact Romanesque.

The early medieval cathedral was razed to the ground: so what is the source for the idea that the Oratory was originally an ancient mithraeum? Certainly, the long, narrow cave-like appearance with benches circling the perimeter is quite suggestive in this respect.[5] Only in 1969 did Mazzolani definitively clarify that "in the points where the plaster is missing, the wall structure appears clearly medieval" (Mazzolani 1969, p. 77). The entire constructed elevation is therefore medieval, contextual to the Romanesque phase. There is, however, confirmation of "some volumetric precedence" in the area of the oratory and beyond, moving towards the façade (Fiorani 2001, p. 11; Piacentini 2006, p. 141). Is this therefore one of those cases in which "paintings are used to transform a pre-existing structure into a church" (Kessler 2001, p. 93)? If so, what kind of pre-existence could we be talking about, and when?

During the paving of the road between Porta Santa Maria and Piazza Innocenzo III (October 2003–February 2004), la Soprintendenza per i beni archeologici conducted partial excavations in the cathedral area. The archeological findings confirm a complex and stratified urban context with continuity of use of the entire acropolis area from the Italic to the medieval period.[6] Investigating the early medieval phases of the cathedral, Mengarelli focused on two points of the basilica in order to arrive at a hypothetical chronology of interventions in the area where evidence remains (Mengarelli 2006, pp. 69–80; Mengarelli 2014, pp. 103–31). He concludes that at a certain moment:

> "The complex of Roman structures, executed in opus quadrata, was exploited or, in any case, deconstructed, [and that this moment] can be inserted chronologically between the moment of change in the topographical structure of the Roman era, which seems to have taken place between the late antique phase and the early Middle Ages, and the Romanesque construction of the Cathedral". (Mengarelli 2006, p. 72)

We can envision the possibility of pre-existing structures in the level below the Oratory (not yet investigated). Indeed, it is plausible that the portion of the site on which the Oratory stands was once part of the terracing of the ancient temple complex, which may have been altered in the early Middle Ages.[7] Thus, save for the elevation of the perimeter walls and the vault, we can speak of the general pre-existence of a structure that included the area of the present Oratory. This space was created almost as a byproduct of the construction of the Romanesque building, which, being larger than the previous church on the site, incorporated pre-existing structures in its vicinity. This hypothesis receives further support from events of November 1998, when the left side of the cathedral façade was waterproofed and the threshold steps were removed. A partially collapsed underlying vaulted room was discovered, completely filled in with loose soil. Equal in width to Becket's Oratory, its length reaches flush with the bell tower (Ferroni 1998; Mengarelli 2001). Originally, these two chambers comprised a single space, one now divided by a septum that forms the back wall of the oratory. It was therefore possible to investigate the masonry phases of the "other side" of the back wall, determining its stratigraphic units. The upper two portions are contextual to the Romanesque phase; the older lower portion presents ashlars in ancient opus quadratum, repurposed in situ in the early Middle Ages (Mengarelli 2006, pp. 73–74). However, as with the exterior on the south-west side of the building, during the construction of the Romanesque building the same pre-existing blocks were reused in this area. Additionally, according to the same solution adopted during the early Middle Ages, the Romanesque builders decided to use them as a base to close off the Oratory and consolidate the façade above.

It is now clear that the Oratory's spatial layout was defined at the time of the construction of the Romanesque cathedral, 1072–1105. The pre-existences are limited to the reuse of ancient ashlars to form the base unit of the back wall of the oratory. In terms of the general plan of the cathedral, the Oratory extends beneath the first three bays of the upper basilica. Articulated as a single chamber, it measures 13.80 m in length and 4.42 m in width, occupying a large portion of the south porticoed flank of the cathedral and facing the Sacco valley and Via Major (today Corso Vittorio Emanuele). The plan clearly

shows that it is divided into two modules: the presbytery and the roughly rectangular nave. There is a natural difference in level, emphasised by travertine blocks, some of which have been placed as if to create a step and others to mark a barrier distinguishing the two areas. The roof is effectively a very irregular barrel vault, with a flattened arch. A long tuff rock bank hewn as a seat runs the whole length of the Oratory, interrupted only at the junction with the left side by the presence of a block of frescoed travertine. The long seat is decorated with red and blue squares within yellow bands. Leaning against the back wall is the painted parallelepiped altar.

In conclusion, when the Romanesque cathedral was built, earlier structures were also incorporated. This may have resulted in the long rectangular room that included the oratory and reached up to the line of the upper staircase (opposite the façade). At a later stage, making use of the previous walls, it was decided to reinforce the line below the façade and thus divide the space of the oratory. The definition of this dynamic is important in order to understand how such an anomalous spatial pattern was generated, which would influence the layout of the medieval paintings. Not only that, but this reconstruction provides a precise picture of the situation in which Pope Alexander III found himself when he hastily dedicated a place of worship to the English saint he had elected a few days earlier in Segni.

## 5. The Decorative Phases; Reading Path; Painting Technique

The reuse and conversion of the room for Becket's cult, as I see it, by Pope Alexander III, are also confirmed by the presence of a layer of paint below the fully medieval layer. Although the sources on the cathedral are silent, it was possible to discover and map splashes of red ochre spread homogeneously over the whole elevation (with the exception of the vault). This proto-artistic (anthropological) evidence has never been analysed. Those found to date comprise drawings that are low in quality, either with an embryonic quality or patently unfinished: (1) a chipped bottom layer in the area at the entrance near the left bench; (2) a painted face, originally part of a larger scene, in the area below the painting of *Saints Scholastica*, *Nicholas*, *and Benedict*; (3) traces of an earlier painted layer in the upper right-hand edge of the scene of the *Presentation in the Temple*; (4) evidence of an earlier polychrome layer under the veil in the first scene of the life of St. Thomas Becket; and (5) the painting of the altar and the corresponding bovine figure (Figure 3).

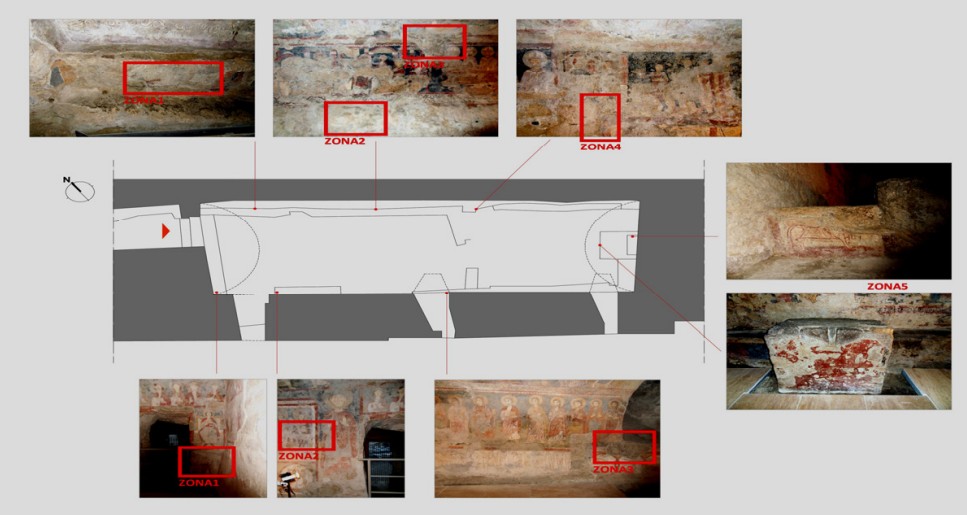

**Figure 3.** Anagni, Cathedral. Oratory of St. Thomas Becket. Previous layer sketches @Arch. A. Felici.

In terms of technique, quality and subject matter, there seems to be no more than a general frequentation or some kind of project that never evolved into an artistic endeavour. It is certain that this phase can be dated before 1173, i.e., before the creation of the frescoes

in which St Thomas Becket appears as a saint. The year 1173 is therefore central to both the cult and the history of the oratory. Within a few days of Becket's canonisation in Segni on 21 February 1173, Alexander III and the Curia moved to Anagni. The only space available to the pope for an immediate artistic initiative is the chamber that becomes the Oratory, a chamber that was already spatially defined and frequented (although for what purpose we cannot say). This part of the crypt levels of the Cathedral offered an ideal location for the insertion of the new cult into a "system of exaltation of the priest and the bishop", a subject that also becomes the main focus of medieval frescoes in the adjoining hall crypt of St. Magnus. The dedication of the chamber to St. Thomas Becket and the ornamentation of the space with paintings related to his story are both more than appropriate for the Cathedral and the town of Anagni itself, which was acting as a papal seat.

The theme of the celebration of Thomas Becket as the most recent example of Priest and Martyr is constructed entirely from an exquisitely ecclesiological perspective. Salvation is eternal not in earthly rulers who pursue only worldly interests, but in the ministers of God who follow the example of Christ in the dimension of Priest and Peter. The scenes of Becket's last moments and martyrdom on the altar of Canterbury Cathedral are part of an extensive iconographic programme. The correct reading of the themes, their arrangement and the stages of their realisation are fundamental steps in determining a correct internal chronology and the meaning of the whole cycle.

Kessler includes these paintings among the "twelve surviving versions, from the twelfth through the fourteenth centuries attesting to the special importance of the Early Christian programme" (Kessler 2002, p. 54). He recognises "a desire to associate the oratory with the Church of Rome [...] most evident in the nave of the chapel, where the cycles of the Old and New Testaments recall the frescoes of St. Peter's, both for the entire programme and in the individual episodes" (Kessler 2001, p. 93). Like any 'medieval copy' of early Christian Roman basilicas, the Oratory expresses a degree of originality in terms of selection and arrangement.

As noted above, the entire surface of the oratory is covered with paintings that collectively amount to roughly 180 square meters of images. Executed in two distinct campaigns with several sub-phases, the paintings present the following macro-themes: on the vault, stories from the *Old Testament*; along the north-east wall, stories from the *Infancy of Christ* (*Annunciation*; *Visitation*; *Announcement to the Shepherds and Nativity*; *Presentation in the Temple*) followed by a panel with remains of *three figures of Saints* (including *Saints Nicholas* and *Scholastica*). The lower registers include a vinescroll motif and, on the protruding part of the wall, a faux-veil decoration. This latter is mirrored on the opposite wall.

This type of decoration is only interrupted at the edge of the presbytery area, where a short section presents a decorative scheme structured as four circles bordered by a circular crown and possibly containing heraldic motifs. The cycle dedicated to the last moments of the life of Saint Thomas Becket begins in the corresponding part of the upper section of the wall. It should comprise, in order: *Henry II and the corruption of members of Canterbury Monastery*; *Accomplice in the Murder/Assassin corrupted by the Devil*; *Murder in the Cathedral*; *Funeral celebration*. The rear wall presents a seven-figure decoration in imitation of the conch of an apse: *Christ seated on a central throne flanked by the Virgin and St. Thomas Becket between two saints*. The vault of the presbytery area is decorated with a circular composition which can be recognised as the *Spiritual Heaven*, inhabited by angels and saints. Beyond the splay of the first window, above the continuous veil decoration, a row of *Twelve Apostles* dressed in tunic and pallium move towards the altar; they stand against a plain blue background without any further determination of place. The vault above the nave shows a figure of *Christ in a mandorla supported by angels*. The next section of wall presents portraits: in isolated panels, although shown seated and facing each other, are *The Sainted Popes Sylvester and Gregory*; further to the right, in the same space are the figures of *The Sainted Pope Remigius* and *Saint Leonard*; finally, three Benedictine monk saints: *Benedict*, *Maurus*, *Dominic of Cuculle*. Below these seven saints, the decoration abandons the theme of the

false veil, to accommodate five intersecting clypeus with decorative-heraldic images. Next to the splayed doorway is a giant *Saint Christopher*.

The counter-façade is occupied by an innovative scene of Judgement featuring the wise and foolish virgins presented in accordance with the iconography of the liturgical drama of the Sponsus. The fate of the foolish virgins, who did not know how to humbly wait for Christ, is foreshadowed by their positioning above a great Devil and a scene of the Weighing of the Souls. The wise virgins, who stand next to the doorway, correspond to the personification of *Humilitas*, which in turn mirrors the representation of Pride on the doorpost.

In terms of the articulation of the iconographic programme in relation to the space, the available surfaces on the one hand and the patterns of narrative arrangement on the other must be taken into account. Some more slavishly follow the "double parallel" scheme (Old and New Testament on opposing wall registers). Others present original modifications rooted either in the 'downward spiral' scheme or in the 'ring' scheme with continuous narration on one level.[8]

At Anagni, one wall of the Oratory includes three openings, the other none. This obliges the *concepteur* to arrange the sacred discourse according to a scheme that is *unique*.[9] (Figure 4). The uniqueness is explained by the fact that the narrative also starts and involves the entire vault, which is divided between several registers with non-continuous shots. This gives an effect more like a labyrinth than a spiral. The circular narrative system resumes in the lower part of the walls.

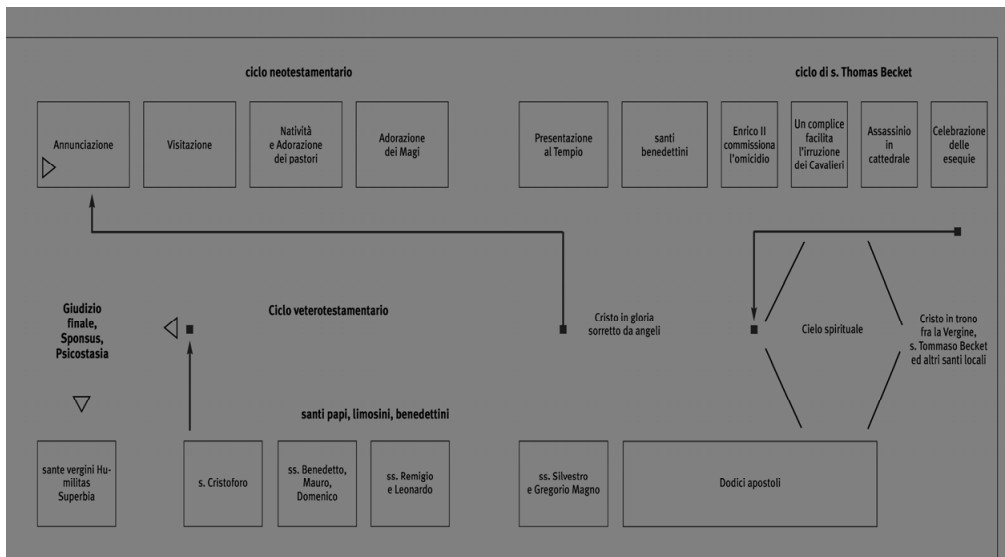

**Figure 4.** Anagni, Cathedral. Oratory of St. Thomas Becket. Icongraphic program and its reading path ©Quattrocchi 2017.

The decision was made to make the most of the largest continuous surface: the vault. The indication of the general directionality is given by the figure of *Christ in a mandorla supported by angels* painted on the portion of the vault adjacent to the border with the presbytery area. Christ also acts as a connector between the Old Testament scenes of the portion covering the nave and the figure of the Spiritual Heaven that surmounts the area in front of the altar.

The vault above the nave is divided into two registers per half-barrel, for a total of four. The narration begins at the doorway in the left half, or north, of the barrel vault. From left to right, the narrative starts from the highest register, moving from the *Creation of the World* to the *Divine Reprimand and Accusation of the Serpent*. It continues in the left half of the barrel, still in the high register, from the *Cherubim guarding the gate of Eden* to *Noah's Ark*. The narrative reprise is always in the right half, or south, starting from the border with

the presbyterial area and flowing from the *Meeting and Offering of Abraham and Melchizedek* to the *Expulsion of Hagar*. Located near the counter-façade, the episodes continue in the lower register of the left-hand half, from *God Appears to Abraham* to *Esau's Return from the Hunt*. The last movement is back to the right. At the height of the central window, one can recognise the figure of *Saint John the Baptist* who, with an unrolled scroll, hints at the above-mentioned and final vision of *Christ in a mandorla supported by angels*.

In contrast to the "downward spiral" of the Old Testament scenes, the New Testament cycle, the panels with saints, and the Becket scenes up to the counter-façade wall are arranged along the walls according to a "ribbon-like and linear" logic. The original *Last Judgement* deserves some attention. Divided into four registers, from the High Heavens above to Hell below, it includes, in its overall composition, both the door and the large window on the south-west wall. When read from right to left, the *wise virgins* and the *foolish virgins* are led, respectively, by *Humilitas* and *Superbia* (this latter on the wall to the right of the window). The alternative is stark: one can choose between the "light" of the Virgins, martyrs and saints or the dark Tartarus of the last register where Satan reigns supreme and the last hope is entrusted to St. Michael as the weigher of souls (Figure 5).

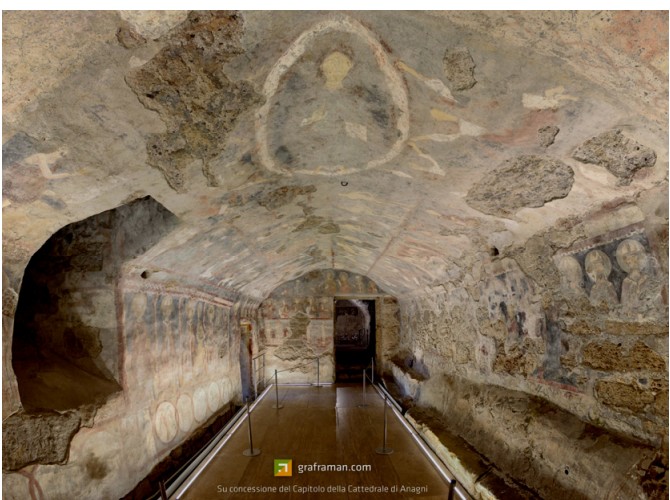

**Figure 5.** Anagni, Cathedral. Oratory of St. Thomas Becket. Vision of the interior © Museo della Cattedrale di Anagni (with kind permission).

Finally, the artistic technique and restoration data. These elements are central to establishing a precise chronology of the phases and interventions. The artistic technique provides a very clear scenario: the entire part of the Becket scenes and the back wall were created in such a hurry and with such a fast technique that the preparatory layer was not even included. The identification of the lines of the plasterwork, which I verified on the basis of the documents, has made it possible to detail the phases of intervention, all of which are roughly close to this first one. As a result of the diagnostic and restoration campaign carried out from 1987 to 2008 by the I. S. C. R., directed by Alessandro Bianchi, it is now possible to identify the different phases of the medieval paintings and to discuss the characteristics of the artistic techniques used. This work will enable us to return to our discussion of the iconography and structure of the paintings from a scientifically informed perspective. The masonry fabric was first partially smoothed by the application of a layer of mortar and then the surface was reinforced by lathing. A second coat of plaster (arriccio), a mixture of coarse sand and poorly ground inclusions, was applied over the lathing. The extremely grainy nature of the arriccio created a very rough base for the frescoes; it should be noted that the arriccio is completely absent beneath the Becket scenes and on the rear wall. The actual frescoing took place in discrete sections of work corresponding to the extent of fresh intonaco (the final plaster layer) laid down at one time (pontata). Two pontata were recognised on each half of the barrel vault, while the walls are divided into

two large horizontal pontata. The vertical joints are not very precise and so they are often easily individuable, allowing for the detailing of the phases.

The second aspect concerns the type and quality of the pigments used. Analysis of blue samples detected the presence of white (perhaps white of lime), a blackish pigment of an undetermined nature, iron oxides and impurities; red samples seemed to consist of red and yellow ochres and fragments of black. The third and final aspect relates more strictly to the painting technique observed using macrophotography: large light brushstrokes were first applied, which were then followed in some places with darker brushstrokes. Technical and environmental reasons are identified as the main factors for degradation, all of which can be attributed to the extreme haste with which the undertaking was carried out.[10]

Approximately ten years after the results of these initial analyses, the director of restoration identified four phases of pictorial intervention in the room (Bianchi 1999). On the basis of these elements and a careful reading of the joints of the painted plasterwork, I propose the following phases: (1a) back wall (*Christ enthroned between la Vergine*, *St Thomas Becket and other saints*); (1b) north wall, *scenes of St Thomas Becket*; (2) presbytery vault; (3) south-west wall (*anonymous saint*) (4a) counter-façade (*Final Judgement*); (4b) south-west wall (*Twelve Apostles*); (4c) nave vault; (4d) north-east wall, *Christological scenes* up to *St. Scholastica*, St. *Benedict*, St. *Nicholas*; (5) south-west wall (*portraits of saints*, *popes*, *Limousins*, *Benedictines*).

Technique and redefinition of the phases define this situation: there was a great urgency to fresco the crypt and, to do so, starting with the Becket scenes. The building site, however, continued in phases that were all very close. These two pieces of evidence fit well with a dating of the whole undertaking close to the canonisation of the Saint (1173) with the Curia and the Pope coming to reside in Anagni less than a month later. Finally, the chronological contiguity of the phases confirms the programmatic intention to insert the Becket scenes and the portrait as a saint in a system of Old-New Testament typological references 'according to the models' of the early Christian basilicas. Replacing the scenes of the Passion of Christ or St Peter with those of Becket was already the concepteur's precise intention.

## 6. The Scenes of St. Thomas Becket

Finally, the last line of research: the iconographic one. What follows is a detailed analysis of the scenes from the cycle of Saint Thomas Becket, which I believe to be the first cycle conceived and produced with this subject (1173–1176). The artistic testimony would even anticipate the literary testimony of the biographies. The four moments in which the story is divided present completely original themes and motifs, some of which have never been attested elsewhere in later examples. It is at Anagni that the Papacy accepts the oral accounts of the murder and allows itself a blatant accusation against King Henry II, seen as the Corruptor/Assassinatee of the Murder. In fact, the case of the Martyrdom scene deserves special attention. Probability, this is the first formulation of a composition of the theme that would become the true manifesto of the cult. It is not unlikely that the direct iconographic source is to be found in the adjacent crypt of San Magnus in the scene of the Martyrdom of St. Magnus. Finally, the back wall presents the first portrait of St Thomas Becket as a saint and in archiepiscopal vestments. The saint himself is to the left of Christ and mirrors the Virgin. This original Deesis is accompanied by a series of local saints. The scenes and the background composition perfectly express Alexander III's intentions: on the one hand to exalt the most up-to-date figure of the prototype of the martyred bishop-saint in favour of his anti-imperial policy; on the other hand, to root and renew the opposition of the local population to the earthly rulers and dominators.

### 6.1. Henry II Commissions the Murder of Archbishop Thomas Becket (?)

The scene is set in a monumental interior. On the left are two characters. The first is dressed in a blue dress and a red conical headdress and appears to be sitting on a throne. The second pushes aside with one hand a curtain separating two rooms, while with the

other hand he holds a bag of white fabric. These actions are directed to the three figures on the right, the first two of whom are lay people and the last a cleric. Their gestures convey a certain animosity and seem to indicate a precise direction. To my knowledge, the scene is unique among all surviving Becket cycles (Figure 6a,b).

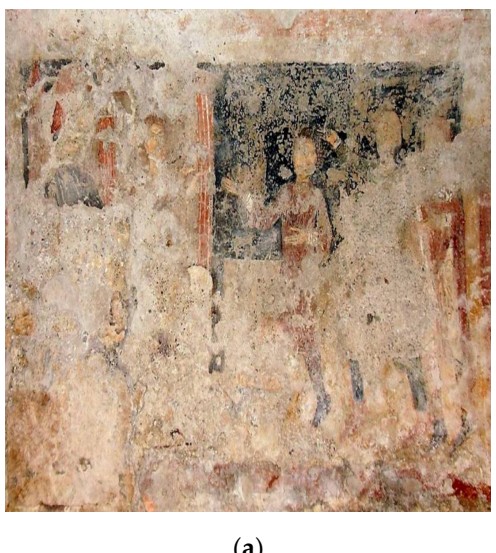
(**a**)

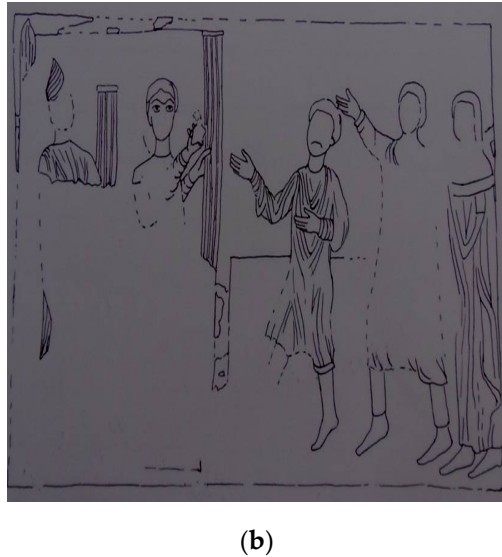
(**b**)

**Figure 6.** Anagni, Cathedral. Oratory of St. Thomas Becket. (**a**) Corruption scene with King Henry II, wall paintings ©author; (**b**) its graphic restitution ©author.

I propose to identify this scene as the act of corruption by Henry II of men implicated in Becket's assassination. This is not to say that they themselves are the assassins, merely that their corruption is the first step to murder. There is no trace of this episode in the sources or in biographical reconstructions. It cannot be ruled out that, if the murder of Thomas Becket is to be blamed directly on the king, the personage on the left is Henry II himself, directing an emissary (see Kessler 2001, p. 95). Such a case appears on bas-reliefs with a Becket subject along the basin of the baptismal font in the church of Lyngsjio in southern Sweden; dated to the late 12th century. In this coeval example, King Henry II, clearly identified by the inscription, personally directs a knight to the next scene, that of the *Assassination of the Archbishop of Canterbury* (Nordstrom 1984, pp. 118–20). Something similar, but certainly more explicit, can be seen in a now detached fresco from the Palazzo dei Trecento in Treviso (around 1260). Becket's short cycle there begins with the scene of King Henry II, the clear instigator of the murder (Cozzi 2008, pp. 75–97).

The scene at Anagni, however, presents a 'more politically correct' scene, no less strong in its accusation. Unlike the cases mentioned above, Henry II does directly instruct an armed personage, but rather uses a court acolyte to hand out money. What is more, the direct recipient of the order is not alone and the assembly also include a bishop. The image almost seems to be an accusation of multiple guilt. Unfortunately it is not possible to establish the identity of the two lay persons present (perhaps they represent the murderous knights in non-military garb) or of the prelate. Nevertheless, it remains that the representation of the offering of money is a public indictment of Henry II (who, it now seems, never gave an explicit order to kill his former chancellor, now archbishop).[11]

*6.2. Accomplice in the Murder/Assassin Corrupted by the Devil*

On the left, the same male figure in a green tunic already encountered in the previous scene can be seen. He extends his arms across a table and seems to open an object in his hands. The adjoining room is a sacred space: we can see on the right an altar with an apse behind it. This man is being corrupted by a demon, seen below the table at right. The demon's eyes and teeth protrude from its mouth; its right shoulder and a

three-fingered hand clutch red spear. The Devil uses an instrument to hand something to the man. On the far right, beyond a curtain, there is evidence of the presence of another character. We can make out the lower part of a nude leg. At the 'collection of the money', the Devil appears (perhaps accompanied by some naked figure as his retinue) (Figure 7). It is precisely the demonic appearance that supports the idea that the corruption of the previous scene is aimed at a criminal action, namely Murder. This idea is present in one of Becket's biographies, which says of the knights who break into the church: "ecce satellites quattro prenominati, cum patre suo Satana a facie Domini egressi" (Benedicto Abbate Pertriburgensi 1854, PL 190, 276A).[12]

Although the scene is unique to date, the introduction of a demonic figure in a "historical tale" is widespread. At Sens, one of the stained-glass windows close in both time and space to that presenting the martyrdom of Becket features a small demon inciting Pontius Pilate at the moment of Christ's judgement. However, even more interesting is what this chartreuse stained-glass window with scenes from Becket's life offers to our view. On axis with the consecration of Thomas Becket as archbishop is a scene commonly interpreted as the discussion between Henry II and the Archbishop of Canterbury at the Council of Northampton. A demonic figure at shoulder height acts as a bad counsellor to the royal, whispering in his ear.[13]

Returning to Anagni, who is the main figure in this scene, the man in green? I suggest that he is one of the future assassins (here not yet in arms) or other facilitator of murder. Perhaps this is a rendering of Ranulf de Broc, the royal custodian of the cathedral chapter's property during Becket's absence. We know that it was Ranulf who directed the knights to an unbarred way into the church via the cloister.

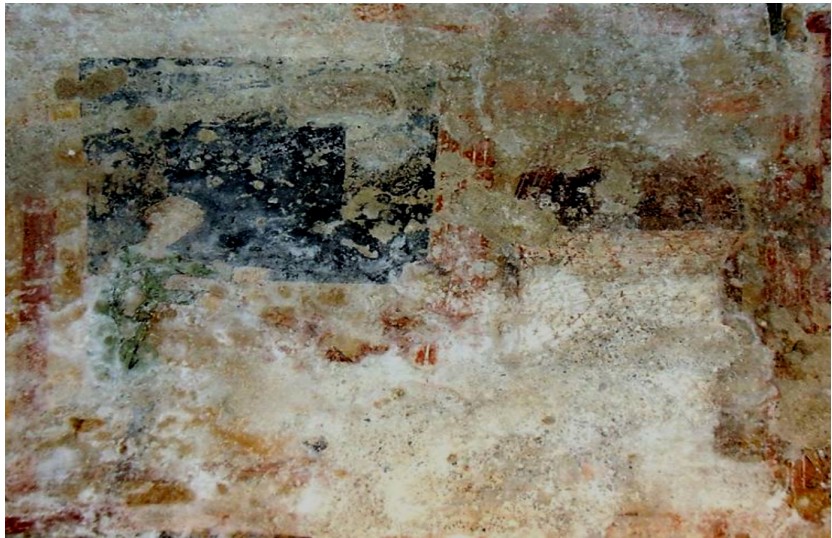

**Figure 7.** Anagni, Cathedral. Oratory of St. Thomas Becket. Accomplice in The Murder/or Assassin corrupted by the Devil, wall paintings ©author.

*6.3. Murder in the Cathedral*

Here, we have the scene of the Raid and the Assassination in the Cathedral (Figure 8a,b).

As can be seen in this drawing of the scene, a total of four knights are involved, in two distinct areas. The Assassination takes place in a well-defined area of the church, recognisable for its roof, columns, and radial apses. The series of blows aimed at Becket are distilled into an image of a shocking rawness. One blow is parried by the cross-bearer Edward Grimm. At the other blows, however, the archbishop's tonsure falls backwards with the force of the sword stroke and, just as will be said in the sources, a fountain of blood gushes from his brain and the same brain hits the ground. No other rendering of the scene carries the same emotional force. Here, violence is instrumentalised by the artists under

Alexander's command: it amplifies the accusation against the king; it justifies immediate sanctity; it exalts the superiority and heroism of bishops over kings.

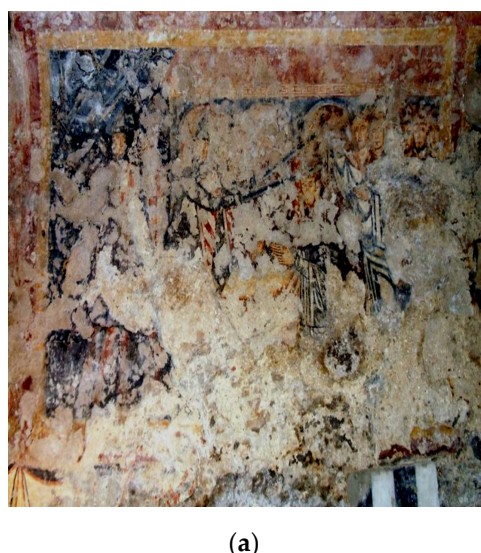
(**a**)

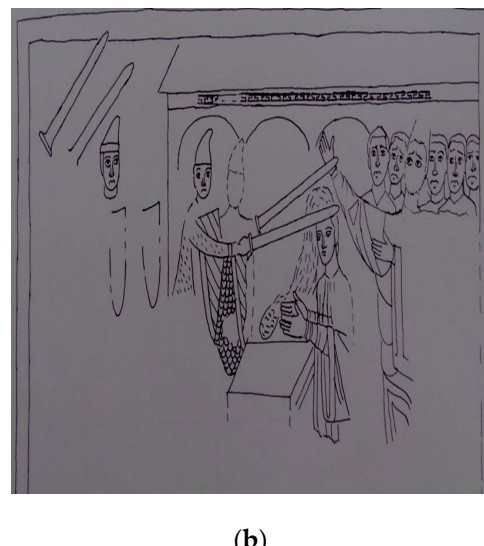
(**b**)

**Figure 8.** Anagni, Cathedral. Oratory of St. Thomas Becket: (**a**) Murder in the Cathedral, wall paintings ©author; (**b**) its graphic restitution © A. Quattrocchi.

The image draws on the most ancient archetypes of sacrifice on the altar. From these iconographic prototypes the painting retains intact its sense of the uncanny, even while updating the subject. The altar is the place of the Priest, but it is also the place of the Offering. In the case of Becket, murdered on an altar, these two meanings overlap: he is both Priest and Offering. His a semantic identification by which the total Christ-type is established. The altar of Canterbury is the new Golgotha. This is also why figural renderings of the setting of the murder become meticulous. Both the body of the Saint and the Holy Place of his sacrifice are central to Becket's cult. Among the most famous paintings with the subject of Murder in the Cathedral is the panel in the Church of Saints John and Paul in Spoleto. Variously dated within a chronological range between 1174 (year of the consecration of the church, a year after Becket's sanctification) and around 1230, it is commonly attributed to the Master of the Spoleto Crucifix.[14] Nilgen (1992, p. 289) recognises in the character who parries the blow for Becket the same Edward Grim (whose severed hand hangs from his wrist). Grim is also present in one of the rare cycles dedicated to the story of the Archbishop of Canterbury: in the apse of the extreme southern arm of the transept of Santa Maria a Terrassa, in the Vallès Occidental (near Barcelona) (Sánchez Márquez 2020, 2021a, 2021b). In Anagni, the iconographic memory of the Assassination of St Thomas Becket is also preserved in the Cathedral Treasury. On display here is a reliquary-box in Limousin enamel[15] containing relics and decorated with subjects from the life of Saint Thomas Becket.[16] The extraordinary diffusion of the Assassination scene throughout Europe is largely due to this category of objects, the production of which is closely linked to the Plantagenet dynasty.[17] The altar and two assassins appear in this example. The excitement of the angular poses is counterpointed by the figure of Becket, hieratically resigned to the event of his impending death.

In addition to the reliquary case, the Cathedral Treasury preserves the memory of the archbishop's martyrdom through a rare artistic object. This is a mitre decorated with the iconography of the Martyrdom of Saint Thomas Becket. The *opus anglicanum* workmanship, together with the historical coincidences, make the provenance of the piece to England plausible.[18] Considered to be one of the oldest examples, the miter of Anagni belongs to the same category of vestments with the same subject matter and technique found in places where the cult spread and took root early on (from Sens to Tarragona).[19] Beyond the

differences, the most relevant cultural fact is, without doubt, the deep-rooted presence of the cult of Thomas Becket in the diocese of Anagni. It is likely that these objects were gifts to the cathedral chapter from some person from England or Normandy. From a broader point of view, the scene of the Murder of Saint Thomas Becket relates to the ancient theme of death on the altar. In the search for the closest hagiographic models, I recall Saint Savinio, bishop of Sens (the city where Thomas spent years in exile) killed with an axe and sword, and Saint Narciso, bishop of Girona, who also died during the celebration of mass.

The most emblematic historical antecedent is undoubtedly that of Saint Elfego, one of Thomas Becket's most distinguished predecessors to the See of Canterbury (1006–1012). During the Danish invasion, he was taken prisoner; because he refused to pay his ransom with the proceeds of his community's alms, he was beaten to death with the bones of an ox served at the invaders' banquet and finally decapitated with an axe on the altar (Eadmero, *Historia novorum in Anglia*, pp. 31–32). Elevated to the honours of the altar in 1078 through the strenuous efforts of Archbishop Lanfranc of Canterbury (in whose curia Becket officiated), Elfego was the first English bishop to die a violent death (Fletcher 2003, p. 78). It is significant, therefore, that in the Christmas sermon of 1170, already foreseeing his own fate, Thomas Becket alludes to the martyrdom of St Elfego in Canterbury Cathedral, stating that another would soon come in the same place (Duggan 2004, p. 202).

However, what was the direct iconographic model? If these paintings are indeed the first artistic testimony, what other image could have inspired its composition?

I believe the easiest and most similar iconographic model to find is in the adjoining crypt. The scene of the Martyrdom of St. Magnus offers the perfect model just a few steps away from the new room to be frescoed. Episodes of the Death and Translation of St. Magnus had been depicted ca. 1105 by the artist known as the First Master. The seven episodes occupy the entire central sector of the walls near the central apse.[20] Among these, on the left-hand side of the apse, is the moment when, after asking to pray for the last time, St. Magnus dies at his altar. Despite finding him dead, the persecutors sent by the Emperor of Decius decide to behead him (Figure 9). In its compositional aspect, the St. Magnus scene is the most direct model for Becket's Assassination[21] Both martyrdom scenes take place within sacred architecture, indeed on the altar itself; in each one there are several killers; the killing of St. Magnus' body is done by scalping. The similarity of the events and the physical proximity of the representation must be considered as decisive factors in taking these paintings as a model for the 'Beckettian subject' of the nearby chamber.

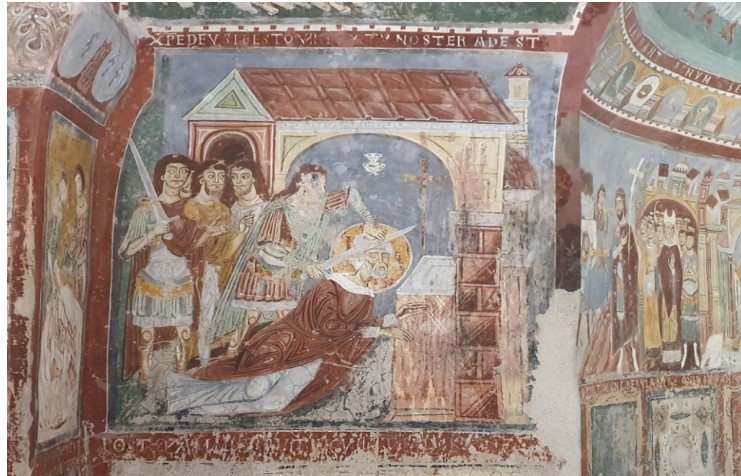

**Figure 9.** Anagni, Cathedral Crypt of St Magnus. Death and Martydrom of St Magnus, wall painting ©author.

### 6.4. Funeral Celebrations

Canterbury Cathedral continues to be present, as a matter of course, in the funeral scene (Figure 10).

The sources specify that the Archbishop's burial became another reason for conflict. The attention of those present gravitates towards a sort of funeral bier on which Becket's lying body is placed. On the far left is a nimbed bishop with miter and vestments, most plausibly Becket himself who, now a saint, with his uncorrupted body welcomes his own soul. In fact, a small figure in Becket's likeness emerges from the corpse, wrapped in white bands. Its hands are clasped in the direction of the saint.

Turning to biographical sources, one finds that the descriptions of these moments are no less graphic than the previous ones. The body of Thomas lying on the ground is hard hit, torn apart. Only after several days was the church was reopened for worship and the coffin shown to the faithful. At this point the collection of the archbishop's blood, still spread across the pavement and already considered to be the testimony of a holy martyr, began (Benedicto Abbate Pertriburgensi 1854, PL 190).[22] The assassins, called "Satan's satellites", tried to prevent the burial of the body near the tombs of Becket's predecessors on the episcopal chair. Only through the resistance of some 'holy men' was it finally possible to lay Becket's body in front of the altars of St. John the Baptist and St. Augustine.[23] At this point, the scene is likely to feature the archbishop's body. Having escaped the danger that the body might be thrown into a mass grave (according to the wishes and intentions of the assassins), it is picked up from the ground, as are the relics. The crowd waiting for this operation consists of the monks present.

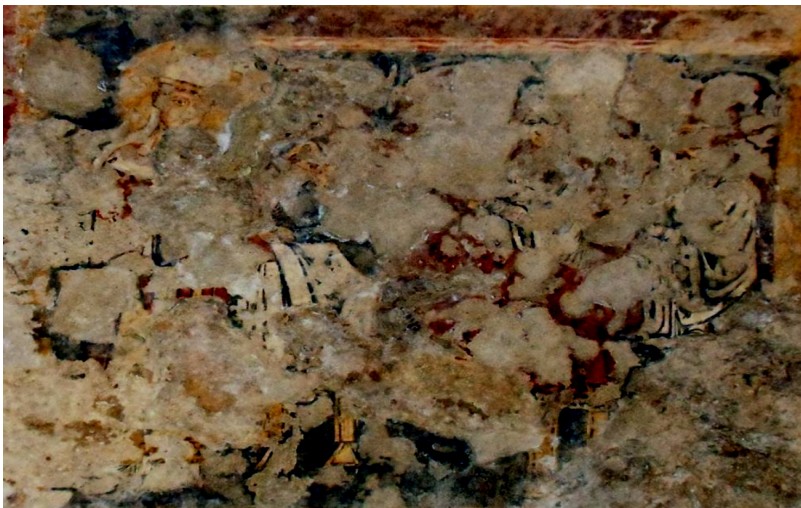

**Figure 10.** Anagni, Cathedral. Oratory of St. Thomas Becket. Funeral scene, wall paintings ©author.

### 6.5. Christ Enthroned between the Virgin, St Thomas Becket and Other Local Saints

The back wall is totally lacking in white preparation, with no underlying layers (like all Thomas Becket scenes).

The composition with seven figures mimics the articulation of an apsidal conch. In the centre, Christ is seated on a gem-studded throne while blessing in the Latin manner and holding a closed codex. On the left, the Virgin Mary stands in the gesture of intercession, leading a procession of two other female saints, whose figures decrease in size according to the availability of space. To the left of Christ is the only figure whose identifying inscription is still partly legible: S. THO/MAS; the conclusion of this label, [...] CHIEPS, visible in 1985, has since faded.[24] The male figure is dressed richly in bishop's robes; he wears a mitre above a rather youthful bearded face with brown hair and holds a crosier in his left hand. He is followed by another bearded, older-looking holy bishop in whom I propose to recognise St. Magnus; the right-hand male group is closed by a tonsured deacon saint in a brown/ochre

robe. Due to the fact that he is the only one not posed in the gesture of intercession, it is assumed that his now damaged right hand could have held an iconographic attribute: he is, perhaps, Saint Leonard (Quattrocchi 2017, pp. 105–9).

In favour of the coexistence of Saint Thomas Becket and Saint Leonard of Limoges there is pictorial evidence that is geographically close to the diocese of Anagni. In the Museum of the Abbey of Casamari (Veroli)[25] there are detached frescoes from the nearby church of Santa Maria del Reggimento (Casamari, Veroli).[26] A lunette depicts the scene of the *Assassination of Becket*, accompanied by three figures of saints. The frescoes, dated to the early 13th century, have never been the subject of systematic study. The similarities between the portrait of Thomas Becket of Anagni and that of Casamari are obvious. Importantly, and in contrast to all other early renderings, only in these two cases does the archbishop wear the mitre and hold the crosier. The episcopal connotation is therefore a clear symptom of the appropriation of the event by the Bishop of Rome, the pope. At Becket's side, we find St. Benedict (?) and St. Leonard, the latter clearly labelled. As at Anagni, his right arm is bent inwards and, together with the left, seems to be holding a sort of chain. As for the remaining female portraits, following this logic they could be the saints Aurelia and Neomisia (Quattrocchi 2017, p. 109).

This is perhaps the most recognisably 'Roman' piece in the whole oratorio; or at least one that aims to create a direct link between the Church of Rome and the chapter of Anagni, where the Curia resides almost permanently.

The compositional structure also suggests a desire for a modern adaptation of the theme of the *Deesis* towards its "actualization" and "extensive local rootedness" (Figure 11). The painting "in this respect is an early example of the variation in apsidal iconography that would become widespread in the 13th century" (Kessler 2001, p. 94). The original 'trimorphic' composition is modulated by keeping Christ in the centre and the Virgin Mary on the left, but substituting St. Thomas Becket for St. John the Baptist. Thus, while Mary reiterates the central concept of the Incarnation, Becket's presence suggests Christ's action on earth and his reification in the ministers of the Church and in the *exemplum of* the saints. Intercession, then, is a choral action in which all the saints particularly linked to Anagni participate, of which the Archbishop of Canterbury is the last in order of time. The substitution of Becket and St. John the Baptist ennobles the sacrifice of the archbishop of Canterbury in three ways. On the one hand, it reaffirms his elevation to the glory of heaven; on the other, it raises that 'new saint' to the same dignity as the martyrs and saints who have long since been canonised; and finally, it roots the cult in a local context with the same function of intercession on behalf of the faithful.[27]

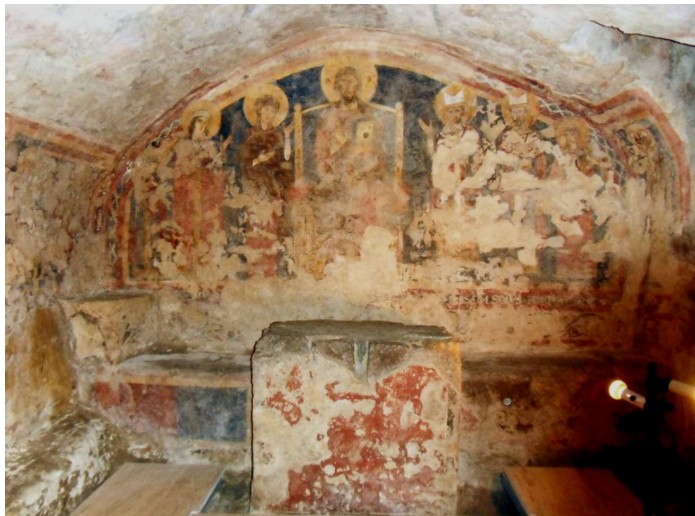

**Figure 11.** Anagni, Cathedral. Oratory of St. Thomas Becket, back wall. Christ enthroned among Virgin Mary, Thomas Becket and other local saints ©author.

Remaining with the central Christ-Virgin composition, the suggestion of and voluntary reference to Rome could be seen in the 'copy' of the most important Roman icons: the Acheropita in the Lateran and the Madonna *Advocata* in Santa Maggiore.[28] Strictly speaking, Becket's portrait at Anagni must therefore be the oldest (or among the oldest). Foreville acknowledges that 'the most venerable effigies of the saint are undoubtedly to be found in a pictorial cycle in a crypt or grotto which extends beneath the cathedral of Anagni and was dedicated to the new martyr immediately after his canonisation' (Foreville 1981, p. 26). She distinguishes between portraits from the Anglo-French and Italian areas. In the first case she recognises a very marked physical type with 'an elongated, often emaciated face, rather globular eyes, an aquiline nose, a slender build' (Foreville 1981, p. 27). The adherence to the information given by the biographers on the subject becomes 'typification' in the case of the portraits from the Italian area; in these the individual features fade away in favour of a 'Norman-type' saint.

It is precisely the Norman presence in Italy that is called upon to justify the attestation of the first images of St. Thomas Becket spread throughout the south of the Peninsula. The one in the apsidal cylinder of Monreale is still commonly believed to be the first monumental portrait (Nilgen 1992, p. 289). Despite the lack of precise sources, the mosaic work is dated to around 1180 [29] (Brodbeck 2013, pp. 271–86). Here, the saint is depicted in full image and blessing, but without the strong episcopal characterisation found in Anagni. After the Norman conquest of Sicily, state and family reasons strengthened relations between the rulers of the island and those of England. Through the Queen Mother, Margaret of Navarre, various people arrived in Sicily from Normandy to assist her in the period of regency for her son. Peter de Blois became chancellor of Sicily and archbishop of Palermo; Richard Palmer was elected bishop of Syracuse at the time of William I, known as Malo (1154–1166); Bartholomew of the Mill became archbishop of Agrigento and his brother Walter canon of Cefalù, as well as tutor to the young William II (Jamison 1992, pp. 301–4).

In 1176, three years after the canonisation of Thomas Becket, Alfano di Camerota, Archbishop of Capua, was sent to England as an envoy to arrange the marriage between William II and Joan Plantagenet, one of Henry II's daughters. With the marriage in 1177, the Hauteville and Plantagenet dynasties were united, further consolidating political relations. These are the circumstances that explain the inclusion of the portrait of St. Thomas of Canterbury among the images of saints in the apse of the cathedral of Monreale, conceived as a dynastic pantheon. A second early portrait of St. Thomas Becket is also connected to these Sicilian-English events, linked to the marriage between Joan Plantagenet and William II. I refer to the enamelled plaque depicting the archbishop in a half-length pose that was inserted into the back valve of the precious Evangelary cover, now in the Treasury of Capua Cathedral, commissioned by Alfano di Camerota and produced in the Norman royal *ergasterion*. In the centre of the valve is the Crucifixion and in the corners are circular plaques with the portraits of the Evangelists and Becket. If Alfonso's mission to England took place in 1176 and the marriage in February 1177, it is very likely that the object dates from just before 1176. In both of these early masterpieces, St. Thomas Becket never appears with a mitre and crosier.

The Oratory would therefore not only house the first monumental cycle dedicated to Thomas Becket, but also the first portrait. There is more: among the earliest representations of his life, this is the first one that insists on the bishop's role. It is a fact that must be taken into account, because it is a strong indicator of a commission linked to the Church as an earthly institution governed by the Vicar of Christ: the bishop of Rome, the Pope. Becket in Anagni is not just a saint; he is a holy bishop of the Church of Rome founded by St Peter. Finally, through the painting on the Oratory's rear wall, St. Thomas Becket also officially joins the list of local saints venerated in the cathedral. For the first time ever—mere months following his consecration into heaven among the saints—he is depicted at Christ's side, mirroring the Virgin: almost a sort of Deesis. The saint of Canterbury has become the new Martyr of the Church of Christ. The body that was previously mangled, outraged

and offered as a Eucharist on the altar now becomes the saint's intact heavenly body.[30] However, this is not enough. In this composition, saints from Anagni appear at the sides. From Canterbury to Anagni and back, Becket's cult and his image is now widespread in the Christian world.

## 7. The Ultimate Message. The Supremacy of the Priest over the King

Within the profound coherence of the iconographic programme, the most original feature is the expansion of the episodes of *Melchizedek and Abraham*: two scenes, one of which was created ad hoc. They are the keystone within which to read the entire decoration and within which to include the uniqueness of the stories of St Thomas Becket. The message of the whole programme, constructed using a coherent typological method, revolves around the theme of the Supremacy of the Priest over the King.[31]

The old scheme of the Investiture Controversy is reproposed; in this second phase, the argument appears even more desperate and violent. The history of the universe and of the humankind is presented as an ongoing drama of each party's choice of which side to entrust its obedience to. All the stories are thus framed in a dualistic perspective between obedience to the pope, as chief of the only institution of divine right, and secular rulers. The alternative is therefore presented in the context of human history, showing the fierce consequences of yielding to evil seduction.[32] Thus the Progenitors, who sinned in Pride by disobeying the divine dictates, are left with nothing but the threat of the Cherub's flaming sword. Thus, in the extended story of *Abel and Cain*, the conflict explodes into a fraternal dimension. The allusion to Becket's biography is immediate. It should not be forgotten that Thomas went from being a close friend of the king, his chancellor and his son's tutor, to a fierce opponent as a defender of the rights of the Church. After the final rupture following the Council of Claredon (1164), he hardened his position even more, to the point of having to leave England for many years.

As in the biblical story, Cain first pays homage to the deity together with Abel, then commits fratricide. The presence of the personification of the *Voice of Abel's Blood*, as well as being one of the dating elements of the cycle, has direct metaphorical and allegorical implications. It is the very blood of the man defined as the 'first martyr' that cries out from earth to heaven. The voice of that appeal is at once the voice of Abel and Becket. The Old Covenant is renewed, therefore, only with a new priesthood, the priesthood according to Melchizedek. Since the time of Philo of Alexandria, passing through a pivotal text such as the Letter to the Hebrews, the foundations of allegorical interpretation have established the Melchisedek-Christ typology. In order to be freed from the datum of heredity of blood (as was Abraham's priesthood), this new Melchisedeck emancipates himself from the historical dimension", surpassing it on the level of duration and spiritual strength. Thus, "all that remains for Abraham is to recognise Melchizedek's superiority, relegating to himself the role of king-conductor who tithes the spoils" (Quattrocchi 2015, pp. 593–606). Melchizedek's superiority, prefiguring that of Christ, extends to his vicar on Peter's Roman seat, going so far as to include St Thomas Becket himself. He is the Melchizedek of modern times to whom, alone, Abraham/Henry II must recognise the priestly identity and supremacy, keeping for himself that of leader. In this way, through obedience, Abraham ensures the life of his son Isaac, who is finally spared by God. Once again, we can read a reference to the clashes between Henry II and Thomas Becket, when the latter refuses to celebrate the coronation rite of the royal boy. There is also a clear admonition to the hope of the continuation of the royal lineage, bound to the submission of a divine recognition of it. It must be said that, from the very first scenes, the theme of the *Sacerdotium* is intertwined with that of sacrifice with a Eucharistic meaning. There are continuous sacrificial offerings, constantly referring to the sacrifice of Christ, the first priest-king who sacrificed himself to redeem humanity. The constant presence of altars and blood pledges iconographically unites the whole cycle to the altar where Archbishop Becket met a bloody death. Even the few Christological scenes are wisely chosen to emphasise the recognition of the only universalistic Kingship on earth: that of the Son of God. Thus,

the *Annunciation* has its place, as the moment in which the divine origin of Christ's earthly experience is illustrated; the *Visitation*, already the first recognition of the supernatural nature of the event; the *Nativity* combined with the *Adoration*, where the community pays homage to the new and true King; the *Presentation in the Temple*, when even the old priestly *auctoritas* recognises the Child. The halting of the story about Christ at this point is no accident. There is a momentary pause in the panel with the founding saints of Benedictine monasticism.[33]

Then, on the same wall, but having crossed the transept area, there is the cycle with the stories of the *Martyrdom of St Thomas Becket*. Rather than avoidance, this is a case of declared substitution of Christological episodes. Even more clearly, the scenes relating to the *Passion* and *Crucifixion of* Christ are replaced with those of the *Passion* and *Sacrifice of* the new English martyr. The visual construction of the Christ-Becket typology makes it possible to enrich the syllogism with other propositions. In fact, by placing the Becket episodes in an area that acts as a "transept", one has a literal "copy" of the positioning of the stories of St Peter in the Vatican basilica (as in other "Romanesque copies") (Kessler 2002, pp. 235–46). At Anagni, the result is a typological chain built from Christ to Peter, from Peter to Thomas Becket. Like Christ (and like Peter, in part), the archbishop of Canterbury is also sold, defrauded and killed. The fact that the iconography proposed in the scenes is largely unpublished and, to date, never found in other examples, would suggest that these are the very first formulations of the theme. This would be supported, as we have seen, also by the painting technique and the history of the painting phases in this area.

## 8. Conclusions. St. Thomas Becket as Propaganda Manifesto of Pope Alexander III

The final nodal question concerns when and by whose will a sacred space in the crypt of Anagni was dedicated to an apparently exogenous cult (see Marazzi 1987, pp. 199–206).

The absolute originality of the scenes would speak in favour of a close temporal proximity between the events and their representation. The representations could have draw directly from oral sources circulating at a time when iconography was not yet standardised. These reasons and others of a specifically historical nature identify the *concepteur* as Pope Alexander III.

As already mentioned, Cardinal Rolando Bandinelli was the personal secretary of his predecessor, Hadrian IV (1154–1159), the only English pope ever to ascend the throne of Peter. Among the last acts of his pontificate, Hadrian IV signed the so-called *pactum anagninum* in the cathedral of Anagni, which anticipated the idea of a Lombard League against Frederick I. On 1 September 1159, Hadrian IV died in Anagni. The struggle against the claims and descents of Frederick I on the peninsular territory was Hadrian's legacy to his successor. Cardinal Rolando Bandinelli, opposed in his nomination by Frederick (and not only), took the throne with the name of Alexander III. From the moment of his election and throughout his pontificate, Alexander III was to face opposition from four antipopes, governing the Church through a schism that lasted at least until 1178. It is not surprising that, among the first acts of his pontificate, in 1160 he excommunicated the Swabian Frederick, thundering from the aisles of Anagni cathedral.

While the city of Anagni increasingly became the 'centre' of European affairs, an extremely bitter conflict was brewing in the English kingdom. In 1164, following the outcome of the Council of Claredon, the already delicate relationship between the Plantagenet King Henry II and the Archbishop of Canterbury Thomas Becket, once the royal chancellor, was definitively severed. The pontiff followed every step of the English political crisis, trying in vain to soften the tone. Reading the epistolary, one can discern that already in 1168, having perceived the extreme difficulty of reconciliation, Alexander had begun to construct a rhetoric of holiness around the English archbishop. These missives are fundamental because they justify a total awareness of the political weight of the affair and testify to the 'construction of sanctity'. Without knowing this 'crescendo', it is difficult to understand the precocity of the cult and artistic testimony of Anagni.

On 9th October 1170, from Anagni, Alexander III wrote a letter to the archbishops Willelm of Sens and Rotrundo of Ruen, who were on a mission to England on behalf of the pope, asking them to oblige the king, on pain of censure, to respect the promise of peace with Becket. The pope says he is burdened by "anxieties and anxieties" explaining the source of his concerns with a formula, full of human affection and ecclesiological ideology, which condenses the use that will be made of this event in the design of the Universal Church for centuries to come:

> *"quae venerabilis frater noster Thomas cantuariensis archiepiscopus pro honore et libertate ecclesiae invicta fortitudine sustinuit, [...] quoties ad memoria nostra deducimus, toties super ejus admirandae virtutis et patientiae con stantia noster anius hilareliscit"*. (Alexander III, Ep. 769, Ad Willelm Senonensem J. Rotrodum Rothomagensem, PL 200, 708)

In 1171, a year after Becket's death, in a letter from Tusculum, he already describes him as a saint who inhabits the highest spheres of heaven: "*[...] cujus anima Deo, sicut credimus, pretioso martyrio dedicata in coelis cum sanctis habitat [...]*" (Alexander III, Ep. 790, Ad Jocium Arch. Tutonensem, PL 200, 727B). The occurrence of a death near a cathedral altar at the hands of armed knights was a dramatic event that soon echoed throughout Christian Europe. Finding himself politically impotent, Alexander III turned a legal crisis into the main topic of his anti-imperial curial propaganda. Forced to move away from Tusculum,[34] the papal curia transferred to Segni, where it remained from early February to 20 March 1173. From St Peter's Church, on 21 February 1172, a just two years after his death, Archbishop Thomas Becket was made a saint by Pope Alexander III. On 12 March, official ratification of the cult was given to the Canterbury chapter, which responded by testifying to the numerous miracles already attributed to Becket's body (Alexander III Romani Pontifici 1855, PL 200, 900–901).[35] A few days after the canonisation, Alexander was the first pontiff to express 'the idea of the martyr's intercession in the remission of sins' (Foreville 1981, p. 28), instituting the 'Jubilee of Becket' for those who went to honour the saint's tomb in Canterbury (Foreville 1981, pp. 13–17). From 21 March 1173, the curia resided in Anagni.[36] As the seat of the curia in years when Rome was insecure for the pontiff, Anagni became the 'centre' of all activities connected with the pontiff. The cult of the Canterbury martyr radiated from here; here this event, only geographically distant, was included and exalted within a 'Roman' scheme of witness and martyrdom of the priest against the king.

A whole series of reports closely connected with the cult and history of the Canterbury See are held in Anagni.[37] The *Chronicon of* Romualdus reports that Henry II sent ambassadors to Anagni to demonstrate his sincere repentance. Perhaps the perpetrators of Becket's murder arrived at the same time: exactly those who appear in the paintings. On 30 December 1173, '*regi in hoc placere credentes*' Hugo de Moreville, Guillelmo de Trachy, Reginal fitzUrsi, Riccardus Brit and Robert de Broc appeared before the pope to repent, prostrating themselves at the pope's feet in Anagni cathedral (Romoaldi II archiepiscopi Salernitani 1866, Chronicon, pp. 438–39). [38]

In the meantime, in England, the issue of the succession of Canterbury bishopric came to the fore. Seeking an ecclesiastical policy more favourable to royal concerns, Henry II designated Richard of Dover, whom he mistakenly believed to be on his side. Richard consequently encountered opposition from the faction of the chapter most loyal to the Church of Rome, which in turn nominated Odo. The result was yet another dangerous dispute. Pope Alexander III received both Richard and ambassadors from Odo's party at Anagni. The judges' reservations were dissolved on 2 April 1174, and on 7 April 1173 the Pope celebrated the consecration of Richard of Dover as Archbishop of Canterbury and Apostolic Representative in the Cathedral of Anagni (Ambrosi De Magistris 1889, II, p. 147).

In June 1174, Henry II, wishing to redeem his image as a Christian and his authority as a king, made a personal pilgrimage to Canterbury, publicly displaying his repentance at the saint's tomb. From October 1174 to 1175, the curia resided in Ferentino, returning to Anagni in 1176 (see Ambrosi De Magistris 1889, II, p. 147). These were the same years of the escalation of Alexander's conflict with Frederick I. It should not be forgotten that, precisely in 1176, the second *pactum anagninum*, signed after the defeat at Legnano, was ratified in Anagni. It became the preliminary act to the Peace of Venice of May 1177. The reconciliation between the Lombard League and Frederick I also had an effect in Rome, where the conflicts between the Commune, the Papacy and the curial members of the families favourable to the imperial side were momentarily quelled. On 12 March 1178 Pope Alexander III entered the city triumphantly. The concordance was short-lived, however; in September of the same year, the Pope and his entourage were found to be residing once again in Anagni.

One year later, in September 1179, a bull of consecration was issued for Anagni cathedral (wrongly judged to be the first ever for the building).[39] On careful reading, the consecration is the normal liturgical conclusion of the translation of holy relics from the altars of the crypt to those of the upper basilica.[40] There is no mention of the Oratory, perhaps because it had already been completed and consecrated. The bull, however, is confirmation of the pope's direct interest as patron in the cathedral.

In 1180, unable to return to Rome, the Pope moved back to Tuscolum, then to Viterbo, and finally to Civita Castellana, where he passed away the following year.

In terms of chronology, the decoration of the Oratory of St. Thomas Becket must certainly be assigned to the will of Alexander III. The saint's canonisation in March 1173 is the *post quem*. The haste with which the work was carried out suggests that the entire first phase, comprising the majority of the paintings, was completed by October 1174, immediately before the institution of worship in the chamber. Motivated by the urgency of finding a place for the new cult, all that was needed was to rebuild a room in the lower part of the cathedral crypt and a quick decorative campaign that at least dealt with the 'most recent subjects'. This date should not come as a surprise. From the moment Alexander III perceived the inevitability of the crisis between archbishop and king, he began to construct a rhetoric of the new martyr of the Church: a new cult, to be fixed forever in new images against internal and external opponents "pro honore et libertate Ecclesiae".

**Funding:** This research received no external funding.

**Institutional Review Board Statement:** Not applicable.

**Informed Consent Statement:** Not applicable.

**Conflicts of Interest:** The author declares no conflict of interest.

## Notes

[1] The staircase exiting the crypt of St. Magnus is perfectly symmetrical, opening onto the third bay of the right-hand nave.

[2] Lastly, see (Thomas Becket: Life, Death and Legacy', International Congress, on line Canterbury Cathedral—University of Kent 28–30 April 2021).

[3] The restoration of the room began with a series of diagnostic investigations in 1985 and was finally completed in 2008. The work was directed by Dr. Alessandro Bianchi who, years ago, kindly gave me permission to view the still unpublished material deposited in the archives of the Istituto Superiore Centrale per il Restauro di Roma (henceforth ISCR). In 2002, the Soprintendenza per i beni ambientali ed architettonici del Lazio, together with a number of scholars from various disciplines, began a thorough restoration of the entire cathedral, both the wall surfaces and the interior of the upper basilica. The results of the work were published in 2006 in a special volume of the *Bollettino d'arte* (Palandri 2006, pp. 1–30).

[4] For the reading and analysis of architectural sites in the cathedral, see (Fiorani 2001, pp. 9–26, 2006-81-89; Urcioli 2006, pp. 187–225).

[5] See (Taggi 1888, pp. 32–33). The idea is supported by the presence of a square ashlar block, reused in the façade of the age of Bishop Peter of Salaerno, where the inscription in capital letters *SEPON* (an abbreviation, according to Taggi, of *serapeion*) is still legible. It was subsequently imagined to have been an ancient prison. See (Zappasodi 1907, I, pp. 116, 122).

6　Among the areas encompassed in the survey is that which extends along the south flank of the church, directly on piazza Innocenzo III. The room dedicated to Thomas Becket naturally faces onto this area. For an account of the excavation, see (Gatti 2006, pp. 49–51).

7　Palandri (2006, pp. 7–8) proposes a reconstruction of the terracing, framing the layout of the south-west flank "as a massive basement and podium in itself […]. In fact, the most important continuity solutions along the slope of the hill are compensated and regularised by the building's own substructures, which are also ordinarily structured as volumes, partly porticoed and therefore historically and logically destined for a distributive-spatial use, and not only for the static-structural purpose of containing the thrust of the soil of the different levels of terracing superimposed on them".

8　The 11th–12th-century cycles dependent on the tradition of St Peter's in the Vatican and St Paul's Outside the Walls unfold according to two possibilities. Among those with a double parallel, see San Bendetto in Piscinula (Rome); San Pietro in Ferentillo (Umbria); San Nicola in Castro dei Volsci (Latium, Frosinone); others in which the double parallel is "hybridised" with a "ring" structure as in Santa Maria delle Grazie in Marcellina (Latium). The second possibility, elaborated around the same period, envisages a "spiral" unfolding criterion starting from the top; thus the wall surfaces are treated as a "continuous" and not "oppositional" support. One of the examples from Lazio is San Giovanni a Porta Latina. This is an "invention" that finds the greatest attestations in the area of central-southern Italy from some passages of Sant'Angelo *in Formis* to the mosaics of Monreale and Palermo (Immonen 2014, pp. 169–97).

9　A similar example in Italy in the Campania region is the oratory of San Nicola near Galluccio. See (Tollo 1994, pp. 85–110).

10　Among the technical ones: poorly mixed mortar; the minimally ground inclusions; the choice of granular and porous material for the lime which favoured increased humidity absorption in the decorated surfaces (and equally fast evaporation). Among the environmental ones: the windows had no screen to the outside for decades.

11　It is widely claimed by historians that the English king's intention was to capture the Archbishop of Canterbury and place him under arrest or in prison. In any case, the four knights responsible for the murder did not have a written order from the king. (Duggan 2004, p. 208).

12　Benedicto Abbate Pertriburgensi, Sancti Thomae cantuariensis arch.-Vita decima-, PL 190, 276A.

13　For a critical account of this detail and its occurrence, see (Jamison 1992, p. 71).

14　This discrepancy insists on two different critical-attributive orientations regarding the interpretation of the Byzantine component in this panel, the mode of which is traced back to the famous Cross from this church and now in Spoleto Cathedral. For an updated review, see (della Valle 2012, pp. 109–17).

15　In general on medieval Limousin production, most recently, see (Caudron 2011, pp. 233–45).

16　Other examples dated to before 1220 (the year in which the relics of St Thomas Becket were moved to Canterbury Cathedral) are in the cathedral of Lucca and in St John Lateran in Rome. Both the Lateran and Anagnine caskets are reproduced more recently (Nilgen 1995, pp. 105–20, Figures 16–17).

17　Between 1167–1169, during the reign of Henri II, the enamellers benefited from the patronage of the Plantagenet family. Henry II, King of England (1154–1189), chose the religious complex of Grandmont Abbey, dependent on Saint-Martirial in Limoges, as the seat of the dynastic *pantheon*. It seems logical that from the reign of Henry II onwards, the Plantagenet dynasty sponsored a production with the subject matter of the last moments of the archbishop's earthly life in order to make amends for the alleged murderous conspiracy. See (Luthan-Hassner 2015).

18　For its variations also in relation to the mobility of workers, see (Privat-Savigny 2014, pp. 81–90).

19　See (Vogt 2010, pp. 117–28) The article, however, makes no mention of the Anagni specimen.

20　For the most recent chronological attribution of the First Master of Anagni's work in the crypt of St Magnus. See (Quattrocchi 2021a, pp. 141–59).

21　For the description of the scene, see (Bianchi 2003, p. 154).

22　Benedicto Abbate Pertriburgensi, Sancti Thomae cantuariensis arch.-Vita decima-, PL 190, 276A: "*Jacente autem adhuc in pavimento in sanguine alii oculos suos liniebant, alii vasculis allatis, quam poterant, partem subripiebant, alii certatim praecisas veste intigebant: nec sibi quisquam postea visus est satis fuisse beatus, qui non de pretioso illo thesauro quantulamcumque reportasset portionem. Et quidem conturbatis et confusis ominibus, licuit singulis quidquid libuit. Pars autem cruoris, quam ecclesiae dimiserant, in vas mundissimum mundissime collecta in ecclesia reponitur conservanda*".

23　Joanne Decano Salisburiensi-Alano Abbate Tewkesburiensi, Sancti Thomae cantuariensis arch.-Vita quarta e quinta, PL 190, 208B: "*In his omnibus persecutorum non quievit furor dicentium corpus proditoris inter sancots pontifices non esse humandum, sed projiciendum esse in paludem viliorel vel suspendendum patibulo. Unde sancti viri qui aderant vim sibi timentes inferri eum in crypta, ante quam satellites Satanae, qui ad sacrilegia perpetranda convocati fuerant, convinerent, ante altare sancti Joannis Baptistae at sancti Augustini Anglorum apostoli in sarcophago marmoreo sepelierunt*".

24　The abbreviation can be completed as [AR]CHIEP[ISCOPU]S.

25　The museum is currently being reorganised. It is divided into a section with archaeological finds and a picture gallery in which the above-mentioned detached frescos are preserved. See (Alonzi 2001, pp. 351–63).

26　For the history of the small church, founded in the 13th century, see (Parlato and Romano 2001, p. 336).



[27] Kessler (2001, p. 95) explains how this process of "adaptation" of the well-known iconography of the *Deesis* becomes monumental in the cases of the portico of Saints Cosmas and Damian and, in an equally "enlarged" version in San Giorgio in Velabro (Rome). Although these latter cases date from the middle of the 13th century, the intention remains to replace the main actors of intercession with Christ with dedicatory or local saints.

[28] It is worth noting that Latium has the largest number of copies of both the Triptych of the Saviour and the so-called *Tempuli* icon, where the Virgin Mary is depicted in the pose visible at Anagni (Leone 2012). On Roman and Lazio icons, see (Leone 2012). I would like to point out an early case of imitations: the Triptych in the cathedral of Tivoli (Curzi 2019, pp. 217–40).

[29] The foundation dates back to 1174; the final *privilegium* dates from 1188.

[30] For the aspect of the representation of the body in the relationship between materiality and the construction of sanctity. See (Quattrocchi 2019).

[31] I have already illustrated the use of the Old Testament as a 'locus politicus'. See (Quattrocchi Forthcoming). Additionally, recently (Quadri 2020, pp. 143–67).

[32] For this general reading. See (Quattrocchi 2017).

[33] It must be read both as a reification of this Christological inheritance and as a cipher of the "Benedictine vocation of this periphery".

[34] In November 1172, an incursion by troops of Roman barons destroyed part of the castle of Tusculum (a town that had recently become a possession of the Pontiff). Alexander went there, but fearing further attacks decided to move with his curia to the nearby town of Segni.

[35] Alexander III, Ep. 1022, Ad Capitulum Cantuariense, PL 200, 900–901.

[36] From October 1174, a1175 (Ambrosi De Magistris 1889, II, p. 147).

[37] On the Canterbury-Anagni relationship, see (Quattrocchi 2021b).

[38] *"Milites autem illi, que in Christi ponteficem cruentas manus inicere non timuerunt, redeuntes ad cor, et reatum suum et nequicitiam recolentes, ad pedes Alexandri pape humiliter advenerunt, et se reso, se esse impios pubblice proclamentes, penitenciam de commissio scelere postulabant. Quos papa, de admissa iniquicitia vehementer redarguit, et venturum super eos Deus Iuditium, nisi digne pentuissent, comminando promisit. Quibus etiam dedit in mandatism,ut discalciati et cilicis induti, Ierusolimam pergerent"* (Romualdi II Archiepiscopus Salernitani, Chronicon, pp. 438–39).

[39] For the events reconstructed here and the full text of the bull, see (Ambrosi De Magistris 1889, II, p. 154–56).

[40] *"In cuius altari maiori idem dominus reliquia recondidit"* (Ambrosi De Magistris 1889, II, p. 156).

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
