# Peer review of "“Pro Honore et Libertate Ecclesiae Invicta Fortitude Sustinuit”—The Oratory of St Thomas Becket in the Cathedral of Anagni"

_arts, 2021_

Round 1

Reviewer 1 Report

This broad contribution is developed in a clear and rigorous manner and is divided into coherent and exhaustive paragraphs.

Author Response

Thank you very much for your opinion

Reviewer 2 Report

The author argues that Pope Alexander III was the concepteur for the fresco program in the oratory at Anagni and that the four scenes about Becket represent papal political ideology related to the Investiture Controversy. Based on a previous monograph on the frescoes by the author, it is asserted here that these frescoes at Anagni are the earliest representations of Becket’s life (1173-76) and they appeared here because Pope Alexander set up Becket’s cult as part of his anti-imperial agenda. This idea is compelling and interesting! The article presents a so-called “vertical” investigation (p. 3) to explain the significance of Becket for Alexander, but this is where I lose track of the overall organization of the argument.

Because of my confusion by p. 3 about the overall thesis, I suggest that the thesis statement needs to appear upfront on the first page. Part 1 on p. 1 is pure visual description, but it would serve the reader better to have this integrated with a basic thesis statement. Where is the argument headed? The second section on Materials and Results attempts to address a thesis, but it is still hard to find. The presentation of questions at the beginning lacks refinement and reads a bit like an undergraduate paper. Also, these questions don’t lead the reader to any sense of what the main argument is. The author mentions a previous monograph on this oratory and the lack of scholarship but fails to mention how the focus of this new article is any different from the monograph. In other words, if we already have the monograph, why is this article necessary? Is it just a recap? This issue is still unclear to me.

When we come to section 3 on p. 3, the sentence, “Following a logic that flows from the outside to the inside, the first step is to understand…” needs greater clarity in terms of intention. The first step of what? Interpreting the political ideology of the frescoes? Redating the scenes? Determining the broader relevance of the earliest cycle of Becket? What are we doing here SPECIFICALLY? Is the hypothesis that the chamber pre-existed its dedication to Becket new? (No, so not the thesis, right?). What about the general volumetrics? How is that a thesis? We are then to learn about the programme in general, the reading path and the artistic techniques used…but to what end? After this point the paragraph about Rome is outright confusing. This section needs ironing out, for it seems to me the author has it in mind to provide a new political reading of the iconographic program at Anagni by considering it in light of earlier papal programs in Rome, the center of papal authority over time. Is that the thesis in a nutshell? If so, that should be what we see at the beginning on p. 1. It could be reiterated in section 3 as the main purpose.

After reading the entire article, these are the main points I think form the thesis:

  • The first part explains why the author redates the frescoes to Alexander’s period, using new archaeological findings about the original building and by examining artistic technique.
  • The second part examines the iconographic program at the Oratory in light of papal programs at OSP and Old St. Pauls, etc., but also by looking at Alexander’s politics and Norman influence.
  • The program expresses the importance of the Becket’s episcopal role within the hierarchy of Church in heaven and on earth. This message is in line with Alexander’s politics/context.

I offer this as a springboard for the author to use to better formulate the thesis at the beginning.

The author and the editors should also evaluate the topic sentences and transitions between paragraphs throughout. In many places I see a basic statement of fact, such as the very first sentence, but no reason for the inclusion of the sentence in the overall paper. For example, another sentence on p. 1, “The current medieval decoration, dated between the twelfth and thirteenth centuries, extends over 180 square meters,” has no purpose for the argument. So what? This happens in many places where I see visual description for the sake of description but with no overall goal of moving the argument along. Another example of this occurs at the last paragraph on p.5 where we just have description without any justification. The paragraph never transitions in any way to the next section. On p. 10 at the beginning of section 6 we have a few lines in a short paragraph that tell us the scenes are unique, but one would love to know how they are unique upfront. What is the new argument here about them? How is this view new? In the section marked 6.4 on p.14 we get this intro sentence, “Canterbury Cathedral continues to be present, as a matter of course, in the funeral scene,” but I don’t see where Canterbury Cath was specifically referenced in the earlier section? This is not much of an introduction to the new section of 6.4 either. We need a better topic sentence/transition to section 6.5 as well.

This article has many good ideas and I think it is worth publishing, but the thesis needs to be ironed out, the transitions and topic sentences need sorting, and the language needs refining. I realize the original language was Italian and so the editors should keep this in mind moving forward. I also notice some minor mistakes in the text that need revision, which I detail below.

Abstract:

The date in the abstract is ordered in the American system (October 9, 1170), but then dates appear in the European system (21st February 1173). Pick one throughout the text.

Sentence “Sensed Becket’s looming tragic fate…” should read “Sensing”; rest of this sentence “began slowly to encircle the archbishop” – this is unclear/awkward.

Suggest “Emperor Frederick I” for those unfamiliar with the conflict and to indicate Regnum/Sacerdotium upfront

“faithful Anagni” sounds like a reference to a dog, but not sure it works here

“to elaborate a powerful speech in images” = unclear/awkward

Do we need “grandiose”?

“program imitating that of St. Peter’s in the Vatican”…program imitating what specifically in the Vatican? The EC frescoes of the nave of OSP…

“the new martyr on the altar of the Church” – can this be explained a bit more here? As worded, it sounds like a sacrifice made by the papacy.

  1. 1 Intro – This visual description could remain the same with an additional paragraph with a thesis statement at the end, but it would be ever so much more elegant if the entire section were integrated into unified argument. Could the first 2 sentences be connected and made into a topic sentence? Who was St. Magnus? (Some scholars compare the Magnus program to the Becket program, so it seems important to briefly mention here).

The word ‘vestibule’ appears here in a single quotation, but I note that this varies throughout the paper with words in double quotation. I’d double check that there is a good reason for that. I think I saw a place where the word was italicized as well.

                Second paragraph – The is the first word of three consecutive sentences. This happens elsewhere on p.4, 2nd paragraph. Vary the word “The” at the beginning of sentences for the sake of interest.

  1. 3 references – I’m not sure I’m familiar with this system of using footnotes together with references in parentheses. Suggest just using footnotes to keep things simple unless this is a European practice in publishing?

The author mentions Kessler and Moretti, but I don’t see Costanza Cipollara and Veronika Decker (2013). Seems relevant as a footnote, particularly as it does not consider Alex the concepteur.

  1. 4 in Section 4; 3rd sentence – It is therefore necessary to begin..what? What is the main point of this section? To argue that the oratory was originally a mithraeum? Lead the reader with better signposts here.

“Building commenced” – suggest His reconstruction commenced?

  1. 4, 3rd paragraph, last sentence – “Let us stop at this date…” Why? What are we even looking to prove at this point? That the earlier construction was…???
  2. 5 “It is now clear that the Oratory’s spatial layout was defined at the time of the construction…” hint at this earlier on p. 1 and maybe at the beginning of this section. Otherwise, I’m not sure why I’m reading this section or how it fits into your thesis about political iconography. This redating seems core to your thesis, no? As it is presented in this section, the redating seems like general background information with no real purpose. Signpost!
  3. 6 Section 5 – We need a paragraph of introduction to this section and why it is here. What will you argue about these decorative phases and why is your thesis new/interesting? Then you can continue with the description

second paragraph here – this begins with a bit of important historical information about Becket’s ties to Alex at Anagni, which you hint at in the abstract, but never really state outright in the intro on p. 1. Would be good to see this historical link earlier.

Third paragraph here – We need a transition from the previous paragraph/topic sentence here. Lead the reader…

  1. 7 the first quotation by Kessler should also be in double quotation marks.
  2. 8 first paragraph, first sentence: “The decision was made to make the most of the largest continuous surface: the vault.” Why does the reader care about this? Topic sentence. Same deal for the next paragraph. I have no idea at this point what I’m supposed to learn from your description.

Last paragraph – I’m confused about the “downward spiral” here when you mentioned at the bottom of p. 7 that Anagni was arranged in a unique way, presumably without a spiral configuration? Clarify this. Speaking of spirals – you may be interested in the new work by Barbara Franze on spirals and the liturgy of church consecration…

  1. 9 explain why the reader wants to know all this information about artistic technique. How does it help advance your argument? I’m not saying it isn’t relevant information, but I don’t know how it fits into your thesis. Might you state something at the top of p. 9 about how you will propose new dating phases that will justify your dating to Alexander’s period? Could you conclude this section with a summary of how your new identification will aid your iconographic interpretation in some way?
  2. 10 title – St appears here without a period, but elsewhere we see St. with a period. Check throughout.
  3. 10 section 6 first paragraph – again…what will you argue in this section about the iconography? How is your reading unique?

Section 6.1 first paragraph – choppy sentences.  “On the left are two characters.” So what?

  1. 11 first paragraph, a “politically more correct” scene, should read a more “politically correct” scene

second sentence here, “…Henry II does directly instructi”???

I wonder if you could make more about the connection between the conical crown of the individual as a king and compare it to other similar crowns? Is this a common way of representing kings? The money bags/devil motif reminds me of Judas, too…

  1. 11 second paragraph, last sentence with Latin phrase – should be translated into English in the note. Same for other Latin quotations.
  2. 12 Love these arguments!
  3. 13 note a space is needed after footnote 19. Look for this elsewhere, too.

Topic sentences needed at top of 6.4 and 6.5 sections.

  1. 16 this section seems meandering, so suggest more signposts/topic sentences related to your thesis. This section will probably be clearer once you have added earlier signposts and your thesis on p. 1
  2. 17 – you get to the Norman presence here to support the first appearance of Thomas images in Anagni. Might you hint this earlier in the thesis in addition to the connection between Alex and Thomas?
  3. 18 – “The iconographic affinities of this portrait with the monumental port in Monreal are total.” What does this mean?

Avoid “masterpieces”

Now we have a new argument about the images at Anagni and the “first portrait” of Becket, which I love. Could you hint at this significance in your intro as well? The role of the bishop part…” The paragraph is rough in terms of language. I would avoid the phrase, “There is more:…” and later “But this is not enough.”

I love the rest of the paper!!

The bibliography needs considerable editing.

Author Response

I sincerely thank you for your attention and time spent on this review. I have accepted almost all the suggestions. I found them appropriate and really improving.

I have put the most significant additions and changes in red.

The system doesn't work, so I'll email you the text